# Antimicrobial Resistance and Virulence Characteristics of *Klebsiella pneumoniae* Isolates in Kenya by Whole-Genome Sequencing

**DOI:** 10.3390/pathogens11050545

**Published:** 2022-05-05

**Authors:** Angela Muraya, Cecilia Kyany’a, Shahiid Kiyaga, Hunter J. Smith, Caleb Kibet, Melissa J. Martin, Josephine Kimani, Lillian Musila

**Affiliations:** 1Department of Biochemistry, Jomo Kenyatta University of Agriculture and Technology, Nairobi P.O. Box 62000-00200, Kenya; angelimuraya@gmail.com (A.M.); kipkuruis@hotmail.com (C.K.); jkimani@jkuat.ac.ke (J.K.); 2United States Army Medical Research Directorate-Africa, Village Market, Nairobi P.O. Box 606-00621, Kenya; cc.katunge@gmail.com (C.K.); hunterjsmith11@gmail.com (H.J.S.); 3Kenya Medical Research Institute, Nairobi P.O. Box 54840-00200, Kenya; 4Department of Immunology and Molecular Biology, School of Biomedical Sciences, College of Health Sciences, Makerere University, Kampala P.O. Box 7072, Uganda; ashakykiyaga91@gmail.com; 5International Center for Insect Physiology and Ecology, Nairobi P.O. Box 30772-00100, Kenya; 6Multidrug-Resistant Organism Repository and Surveillance Network (MRSN), Walter Reed Army Institute of Research, Silver Spring, MD 20910, USA; melissa.j.martin28.ctr@mail.mil

**Keywords:** antimicrobial resistance, virulence, whole-genome sequencing, Kenya, *Klebsiella pneumoniae*

## Abstract

*Klebsiella pneumoniae* is a globally significant opportunistic pathogen causing healthcare-associated and community-acquired infections. This study examined the epidemiology and the distribution of resistance and virulence genes in clinical *K. pneumoniae* strains in Kenya. A total of 89 *K. pneumoniae* isolates were collected over six years from five counties in Kenya and were analyzed using whole-genome sequencing and bioinformatics. These isolates were obtained from community-acquired (62/89) and healthcare-associated infections (21/89), and from the hospital environment (6/89). Genetic analysis revealed the presence of *bla_NDM-1_* and *bla_OXA-181_* carbapenemase genes and the *armA* and *rmtF* genes known to confer pan-aminoglycoside resistance. The most abundant extended-spectrum beta-lactamase genes identified were *bla_CTX-M-15_* (36/89), *bla_TEM_* (35/89), and *bla_OXA_* (18/89)_._ In addition, one isolate had a mobile colistin resistance gene (*mcr-8*). Fluoroquinolone resistance-conferring mutations in *gyrA* and *parC* genes were also observed. The most notable virulence factors were those associated with hyper-virulence (*rmpA/A2* and *magA*), yersiniabactin (*ybt*), salmochelin (*iro*), and aerobactin (*iuc* and *iutA*). A total of 38 distinct sequence types were identified, including known global lineages ST14, ST15, ST147, and ST307, and a regional clone ST17 implicated in regional outbreaks. In addition, this study genetically characterized two potential hypervirulent isolates and two community-acquired ST147 high-risk clones that contained carbapenemase genes, yersiniabactin, and other multidrug resistance genes. These results demonstrate that the resistome and virulome of Kenyan clinical and hospital environmental *K. pneumoniae* isolates are diverse. The reservoir of high-risk clones capable of spreading resistance, and virulence factors have the potential to cause unmanageable infection outbreaks with high morbidity and mortality.

## 1. Introduction

*Klebsiella pneumoniae* is a Gram-negative, rod-shaped ubiquitous bacterium that inhabits soil, water, and sewage ecosystems. It is also found on various human body sites and organ systems, including skin, nose, throat, and intestinal tract, as part of the natural microflora [1]. *K. pneumoniae* is a prominent member of the *Klebsiella pneumoniae* species complex that consists of seven species that include *Klebsiella pneumoniae, Klebsiella quasipneumoniae* subsp. *quasipneumoniae*, *Klebsiella quasipneumoniae* subsp. *similipneumoniae, Klebsiella variicola* subsp. *variicola, Klebsiella variicola* subsp. *tropica*, *Klebsiella quasivariicola*, and *Klebsiella africana* [2]. The first four species are commonly associated with human infections such as pneumonia, urinary tract infections, soft tissue and wound infections, septicemia, and pyogenic liver abscesses [3].

The World Health Organization (WHO) declared antimicrobial resistance (AMR) as one of the top 10 most serious global public health threats facing humanity [4]. The WHO lists *K. pneumoniae* as one of the AMR bacteria of concern due to its demonstrated proclivity for developing antimicrobial resistance to many classes of antibiotics such as penicillins, cephalosporins, and quinolones [5,6,7] which are typically used to treat *K. pneumoniae* infections. This resistance is due to both chromosomal-encoded and plasmid-encoded genes. The abundance of AMR genes carried on plasmids and mobile genetic elements have earned *K. pneumoniae* its reputation as a “key trafficker” of AMR genes between *Klebsiella* species and other *Enterobacterales*, illustrating its importance to AMR spread and development [8]. As carbapenems are considered one of the last resort treatments for multidrug-resistant (MDR) *K. pneumoniae*, i.e., isolates resistant to three or more drug classes [9], the global increase in carbapenem resistance [10] presents a threat to public health. The situation is aggravated by the isolation of colistin-resistant *K. pneumoniae* [11] since colistin is used as a last-line antibiotic for treating carbapenem-resistant *K. pneumoniae*. Global MDR high-risk lineages such as ST14, ST15, ST147, ST307, and ST607 have been identified and have spread rapidly across the globe increasing the need for global AMR surveillance. In Kenya, ST15 and ST17 have been reported in Kilifi [6] and ST14 in Nairobi [12].

In Kenya, several studies have indicated an emergence of the clonal spread of MDR *K. pneumoniae* and horizontal transfer of AMR genes [6,12,13,14,15]. Extended-spectrum beta-lactamase (ESBL) producing *K. pneumoniae* isolates have been reported in several locations such as Nairobi, Kisumu, Kisii, Homabay, Migori, Kilifi, and Eldoret [6,13,15]. Increased ESBL-producing *Enterobacterales,* the most prevalent of which in Kenya are CTX-M-15 and TEM, is associated with the increased use of third generation cephalosporins in the early 2000s [10,13,16,17]. To exacerbate the situation, carbapenem use in Kenya is increasingly leading to the emergence of carbapenem-resistant *Enterobacteriaceae* (CRE). In 2011, Poirel, et al., were the first to detect CRE in Kenya from urine samples bearing the New Delhi Metallo-β-lactamase (NDM-1) carbapenemase [12]. In 2017, there were reports of *K. pneumoniae* isolates that encoded *bla_NDM-1_* and *bla_SPM_* carbapenemase genes [15] and isolates recovered from bacteremias bearing the pNDM-MAR-like plasmid backbone that has been shown to carry the *bla_NDM_* gene [6]. More recently, Musila et al. (2021) identified *K. pneumoniae* isolates with the *bla_OXA-181_* carbapenemase gene. In addition, *K. pneumoniae* resistance to aminoglycosides, tetracyclines, chloramphenicol, and fluoroquinolones has also been observed in Kenya, typically associated with ESBL production in MDR [6,10,18]. Aminoglycosides are widely used to treat bacterial infections due to the ease of administration and access in Kenya [19]. Musila et al. (2021) also reported the 16S rRNA methyltransferase genes, *rmtF* and *rmtC*, in *K. pneumoniae* isolates which confer pan-aminoglycoside resistance [18]. Resistance to chloramphenicol and sulfonamides is high in part due to its use as a first-line treatment option for enteric infections such as typhoid and HIV prophylaxis in Kenya [10].

In addition to antimicrobial resistance, *K. pneumoniae* possesses virulence genes that enhance its ability to cause infections, increase cell fitness, and evade the host immune system. *K. pneumoniae* colonizes host cells using adhesins such as fimbriae and pili [20]. The production of a robust capsular polysaccharide confers resistance to host immune cells along with the O-antigen portion of the liposaccharide (LPS). *K. pneumoniae* attacks rival bacteria and eukaryotic cells by injecting potent endotoxins using type VI secretion machinery [21]. Highly virulent *K. pneumoniae* increases the expression of *magA, rmpA,* and *rmpA2* genes linked to the mucoid phenotype for hypervirulence [22]. Other mechanisms include the use of allantoin for a carbon and nitrogen source, the use of efflux pumps to eject antibiotics [23], and the capture of iron molecules from the host cells using siderophores (aerobactin, salmochelin, yersiniabactin) [24]. Although some studies in Kenya have identified *K. pneumoniae* lineages ST11, ST15, and ST17, associated with hospital outbreaks [6,14], and the high-risk MDR ST147 [19], few studies have examined virulence genes in Kenyan *K. pneumoniae* isolates [14].

In Kenya, screening of AMR phenotypes, genes, and sequence typing is done mainly using manual or automatic antimicrobial susceptibility testing (AST) and polymerase chain reaction amplification techniques [12,15,17] These techniques provide limited targeted information and are laborious compared to whole genome sequence data. As such, there is limited information on the STs and extent of AMR genes carried in *K. pneumoniae* in Kenya [25,26]. Thus, this study combined whole genome sequencing analysis with phenotypic antimicrobial susceptibility testing to identify the AMR genes and gene mutations associated with the phenotypic patterns observed and also to conduct in silico sequence typing of known and novel sequence types (STs). 

A clearer understanding of *K. pneumoniae* sequence types (STs) circulating within the community and healthcare systems across Kenya and the diversity and distribution of AMR and virulence genes would facilitate the monitoring and control of high-risk clones that are potentially hypervirulent and/or multidrug-resistant. Given the public health and clinical importance of *K. pneumoniae* and the knowledge gaps in Kenya, this study set out to examine *K. pneumoniae* isolates across Kenya and describe: (1) the sequence types and their distribution across Kenya, (2) the antimicrobial resistance phenotypes, and (3) the AMR and virulence genes present in the *K. pneumoniae* isolates.

## 2. Materials and Methods

### 2.1. Study Site

The bacterial culture, DNA extraction, quantification, and Oxford Nanopore sequencing were conducted in the Kenya Medical Research Institute (KEMRI)—Center for Microbiology Laboratory. Illumina sequencing was performed at the Multidrug-Resistant Organism Repository and Surveillance Network (MRSN), Walter Reed Army Institute of Research (WRAIR). The bioinformatic analysis was conducted using the high-performance computing server at the Department of Emerging Infectious Diseases, USAMRD-Africa in KEMRI, Nairobi.

### 2.2. Study Samples

The study analyzed hospital environmental and clinical isolates of *K. pneumoniae* that were collected between May 2015 and March 2020 from eight hospitals in five counties in Kenya as part of an ongoing antimicrobial resistance surveillance study (KEMRI2767/WRAIR 2089) and an environmental study (KEMRI 3482/WRAIR 2416). The clinical samples were urine, wound swabs, and pus collected from consenting patients with suspected bacterial infections. The environmental samples were collected via swabs of high-touch areas in the participating hospitals. *Klebsiella pneumoniae* identification and antimicrobial susceptibility testing (AST) were performed on the Vitek2^®^ system (bioMérieux, Lyon, France) using the GN-ID and XN05-AST cards. The AST panel consisted of penicillins (piperacillin and ticarcillin/clavulanic acid), monobactam (aztreonam), cephalosporins (cefuroxime, cefuroxime axetil, cefixime, ceftriaxone, and cefepime), carbapenems (meropenem), fluoroquinolones (levofloxacin and moxifloxacin), tetracyclines (tetracycline and minocycline), glycylcycline (tigecycline), phenicol (chloramphenicol), and trimethoprim. The AST results were interpreted according to CLSI guidelines (2018) [27], and isolates were classified as either multidrug-resistant (resistant to three or more drug classes) or non-multidrug resistant and ESBL positive or negative.

### 2.3. DNA Extraction and Sequencing

A total of 49 *K. pneumoniae* isolates were inoculated on Mueller Hinton Agar plates and incubated for 24 h at 37 °C. *Klebsiella pneumoniae* JH930422.1 was included as a positive control. Approximately 14 × 10^8^ cells/mL at OD 600 of overnight bacterial cells were re-suspended in nuclease-free water and centrifuged at 5400× *g* for 10 min to pellet the cells. Total DNA was then extracted using the DNeasy^®^ UltraClean Microbial Kit (QIAGEN Inc., Hilden, Netherlands) according to the manufacturer’s instructions. DNA purity was determined on the Nanodrop One (ThermoFisher Scientific, Waltham, MA, USA) and quantified on the Qubit dsDNA fluorometer (ThermoFisher Scientific, Waltham, MA, USA). For library preparation, the extracted DNA was end-repaired using the NEBNext® UltraII End Repair/dA-Tailing kit (New England Biolabs, Ipswich, MA, USA) using the manufacturer’s instructions. Ligation Sequencing Kit-LSK109 (Oxford Nanopore Technology, Oxford, United Kingdom), EXP-NBD 104 (1–12), and EXP-NBD 114 (13–24) Native Barcoding kits (Oxford Nanopore Technology, Oxford, UK) were used to barcode each DNA sample and adapters added. The DNA library was prepared and loaded onto a FLO-MIN106 R9.4.1 flow cell for sequencing based on the standard Oxford Nanopore Technology (ONT) 1D-sequencing protocol. The sequencing run was launched on the MinKNOW software (v20.10.3, Oxford Nanopore technology, Oxford, UK). A quality control experiment was conducted to evaluate the Nanopore workflow. This was done by sequencing the lambda phage DNA on the FLO-MIN106 R9.4.1 flow cell for 6 h as per the Lambda DNA control experiment protocol, and the sequences were compared with the reference sequence (NC_001416.1). Guppy software (v4.4.2, Oxford Nanopore Technology, Oxford, UK) was used to basecall and trim the barcodes and adapters from both ends of the reads.

The whole genome sequence reads of 40 additional *K. pneumoniae* isolates, sequenced on an Illumina MiSeq platform at MRSN-WRAIR as previously described [19], were included.

### 2.4. De novo Assembly of Raw Reads and Database Querying

The fastQ files of the long reads were filtered to retain only those with a Q-score ≥ 7. The adapters in the short-pair-ended reads were trimmed using Trimmomatic v0.39 [28]. The trimmed Illumina reads were assessed for quality using FastQC v0.11.9 [29] before de-novo assembly using the default Shovill v1.1.0 [30] pipeline settings. Next, the draft assemblies were polished using pilon v1.24 [31]. Finally, the ONT long-reads were de novo assembled using flye assembler v2.8.1 [32] with the plasmid option, followed by one polishing round using medaka v1.3.2 [33]. All polished draft assemblies from Illumina and ONT sequencing were analyzed in the same way. First, the quality was assessed using QUAST [34]. Then, the draft assemblies were queried using the ABRicate v1.0.1 [35] pipeline against CARD [36] to identify *AMR* genes, VFDB [37] to identify virulence factors, and PlasmidFinder [38] to identify plasmids. The assemblies were queried against the Klebsiella MLST database using the command line mlst v2.19 [39] pipeline to determine the sequence types, while the capsule (K) and O types were determined using the Kleborate pipeline [40] against the Kaptive database [41]. Next, a new ybt-typing scheme updated in the Kleborate [40] pipeline was explored to assign allelic profiles to yersiniabactin genes. Finally, a maximum-likelihood phylogenetic tree was generated using Parsnp v1.2 [42] and NC_009648.1 as the reference genome. The tables were created using flextable [43] (R package), while the circular tree and heatmaps were generated using Interactive Tree of Life (iToL) v6.3.2 [44] tree annotator.

## 3. Results

### 3.1. Epidemiological and Clinical Characteristics of K. pneumoniae Isolates

Clinical samples were collected from skin and soft tissue infections (SSTIs) (64%, 57/89), urinary tract infections (UTIs) (29%, 26/89), and the hospital environment (7%, 6/89) from five counties in Kenya: Kisumu (39%, 35/89), Nairobi (26%, 23/89), Kisii (17%, 15/89), Kilifi (10%, 9/89), and Kericho (8%, 7/89). A total of 62 isolates were isolated from community-acquired infections (CAI), while 21 were from healthcare-associated infections (HAI).

Of the four members of the *K. pneumoniae* Complex identified [2], *K. pneumoniae* subsp. *pneumoniae* represented the largest proportion of all isolates at 79% (70/89), with 44 recovered from SSTIs, 20 recovered from UTIs, and 6 from the hospital environment. This phylogroup had the largest number of MDR isolates (39%). The second most represented phylogroup was *K. variicola* subsp. *variicola* (18%, 17/89) with 15 isolates recovered from SSTIs and 2 isolated from UTIs. There was only 1 MDR isolate (kkp059) in this category. The least represented phylogroups were *K. quasipneumoniae* subsp. *quasipneumoniae* (2%, 2/89) and *K. quasipneumoniae* subsp. *similipneumoniae* (1%, 1/89) (Figure 1). Among the isolates from these minor subspecies, kkp022 and kkp034 were isolated from a UTI, while kkp036, the only MDR in this category, was recovered from an SSTI. All *K. quasipneumoniae* subsp. *quasipneumoniae*, *K. quasipneumoniae* subsp. *similipneumoniae*, and *K. variicola* subsp. *variicola,* except for one isolate (kkp078), were from community-acquired infections from different geographical locations (Figure 1). There was no evident geographical clustering of all the phylogroups by county or infection types.

### 3.2. Genomic Characteristics of K. pneumoniae Isolates

Draft genomes were generated from the 89 isolates: 40 via Illumina short-read sequencing and 49 via MinION-based long-read sequencing. The sizes of the draft genomes ranged from 5.2 to 5.9 Mb with an average G + C content of 57.27%, typical of *K. pneumoniae* genomes [45] (Appendix A). The average N50 for the short and long reads was 226,402 and 5,147,285 base pairs, respectively. The isolates had 0–8 plasmid replicons, averaging 3 per isolate. The highest number of plasmid replicons were in genomes kkp001 (8), kkp018 (8), and kkp0e21 (8) (Appendix A), whereas five genomes had no predicted replicons: kkp012, kkp030, kkp070, kkp102, and kkp112. The most abundant plasmid replicon types identified belonged to the Col and Inc family (particularly the IncF type) (Figure 2). The other Inc-like plasmid replicons identified were IncR, IncH, IncX, IncC, IncN, IncM, and IncY. Seven types of Col plasmid replicons, the second most represented type, were identified, dominated by Col(pHAD28) (Figure 2). Other plasmid types identified included pKP1433 and rep_KLEB_VIR in kkp056 in kkp043 genomes, respectively. 

### 3.3. Multilocus Sequence Types and Distribution of K. pneumoniae Isolates

The STs could only be assigned to 57% (51/89) of the isolates. The 37 unassigned isolates were not typeable due to low genome coverage (Appendix A). The multilocus sequence types (STs) of the 52 *K. pneumoniae* isolates were diverse, with 38 different STs identified. STs that were represented by more than one isolate were ST15 (4/52), ST17 (3/52), and ST607 (3/52 each); two isolates each represented the ST14, ST37, ST39, ST48, ST147, and ST307 lineages (Figure 3). The remaining 29 sequence types were represented by a single isolate and were geographically distributed as follows: Kisumu (*n* = 17), ST20, 45, 101, 198, 336, 391, 751, 1927, 2010, 3717, 3397, 3609, 5594, 5595, 5596, 5598, 5599; Nairobi (*n* = 6), ST55, 219, 966, 1786, 3692, 5600; Kisii (*n* = 4),ST25, 711, 3717, 5597; and Kilifi (*n* = 2), ST364 and 5593. Eight isolates had novel allelic profiles assigned and deposited in the *K. pneumoniae* MLST database (https://bigsdb.web.pasteur.fr/) (Table 1). There was no evident clustering of STs by infection type or ESBL and MDR status. Kisumu had the highest number of distinct STs (23), followed by Nairobi (12), Kisii (6), Kilifi (3), and Kericho (2). The high-risk lineages, ST14, ST15, ST17, ST307, and ST607, were concentrated in the biggest cities: Kisumu and Nairobi. Global high-risk strains ST14, ST15, ST307, and ST607 were predominantly from the *K. pneumoniae*.

### 3.4. Antimicrobial Resistance

#### 3.4.1. Phenotypic Resistance Profiles

Antibiotic susceptibility tests (AST) performed on the VITEK2^®^ platform (bioMerieux, Lyon, France) (Appendix A) identified high levels of non-susceptibility to trimethoprim (57%, 51/89), ticarcillin/clavulanate (49%, 44/89), cefuroxime axetil (49%, 44/89), cefuroxime (48%, 43/89), cefixime (48%, 43/89), ceftriaxone (47%, 42/89), cefepime (46%, 41/89), aztreonam (47%, 42/89), tetracycline (37%, 33/89), and minocycline (34%, 31/89); moderate non-susceptibility to levofloxacin (19%, 17/89), moxifloxacin (20%, 18/89), and chloramphenicol (19%, 17/89); and low levels of non-susceptibility to tigecycline (4%, 4/89) and meropenem (2%, 2/89). All isolates were non-susceptible to piperacillin (Figure 4). Among the dataset, 41% (37/89) were classified as MDR, i.e., resistant to three or more drug classes. A large proportion (47%, 42/89) were ESBL-producing isolates (Figure 4), of which 30 isolates were MDR. One isolate, kkp001, was pan-resistant (resistant to all 16 antibiotics tested in the panel).

#### 3.4.2. Genetic Determinants of Resistance

Comparison of the isolates’ phenotypic and genotypic antimicrobial susceptibility results demonstrated high concordance in the trimethoprim, beta-lactam, and tetracycline antibiotic classes, as demonstrated in Figure 4. In particular, there was strong concordance among the *K. pneumoniae* isolates between the presence of a *bla_CTX-M_* gene and non-susceptibility to beta-lactams and cephalosporins (Figure 4), *dfrA* gene with trimethoprim non-susceptibility, and *tet* genes with tetracycline non-susceptibility. Most of the genomes generated from long reads enabled the detection of circularized plasmids bearing AMR genes (Table 2). MDR isolates had more plasmid replicons than the non-MDR isolates which had only 0–3 plasmid replicons (Figure 4). 

*Beta-lactamase resistance genes.* In the beta-lactam antibiotic class, genes for penicillinases, ESBLs, and carbapenemases were identified, summarized in Appendix A. Each phylogroup demonstrated a distinct chromosomal penicillinase (Appendix A), i.e., *K. pneumoniae* contained *bla_SHV_*, *K. quasipneumoniae* contained *bla_OKP_*, and *K. variicola* contained *bla_LEN_*, consistent with previous studies [8]. However, some isolates carried a different additional penicillinase which was observed to have been acquired via a plasmid, e.g., kkp036 (*K. quasipneumoniae*) obtained a *bla_SHV-134_* from an IncN plasmid (Table 2). There was a high diversity of plasmid-mediated ESBL genes in the isolates: *bla_CTX-M-15_* (40%, 36/89), *bla_CTX-M-98_* (1%, 1/89), *bla_CTX-M-14_* (1%, 1/89), *bla_TEM181_* (39%, 35/89), *bla_OXA_* (18/89), and the less common *bla_LAP2_* and *bla_SCO1_*. Two MDR isolates had carbapenemase genes: *bla_NDM1_* (kkp003) and *bla_OXA181_* (kkp001). Isolates kkp013 and kkp063 did not demonstrate any β-lactamase genes, and they were unsurprisingly ESBL-negative and non-MDR. Most isolates possessed at least one and at most six β-lactamase genes (Appendix A). There were 42 ESBL-positive isolates, and 38 of them had a *bla_CTX-M_* co-harbored with 2 or more other types of ESBL genes (Figure 4). 

*Non-beta-lactamase resistance genes.* A previously described [11] colistin-resistant isolate, kkp018, carried an *mcr-8*, a *bla_DHA1_*, and 20 additional AMR genes (Appendix A). A total of 67% of the isolates (60/89) carried aminoglycoside-modifying enzymes (AME) genes, i.e., *ant(3′)* (or *aadA*) encoding aminoglycoside nucleotidyltransferases; *aph(3′)*, *aph(4)*, and *aph(6)* encoding aminoglycoside phosphotransferases; *rmtF* and *armA* encoding16S rRNA methyltransferases; and *aac(3)* and *aac(6′)-Ib-cr* encoding aminoglycoside acetyltransferases (Figure 4, Appendix A). The *aac(6′)-Ib-cr* gene also induces fluoroquinolone resistance. The genes associated with pan-aminoglycoside resistance were found in two isolates: *armA* (kkp018) and *rmtF* (kkp001). The genes for efflux pumps associated with aminoglycoside resistance (*arnT, crcB, acrD, baeR, cpxA*) were well-conserved among all the isolates. In addition, the common MDR efflux pump genes were identified: *LptD, CRP, H-NS, KpnEFGH, acrAB, marA, mdtBC, msbA,* and *ramA*.

Three mechanisms for quinolone resistance were detected. First, plasmid-mediated quinolone resistance (PMQR) genes (*qnrB* or *qnrS1*) were identified in 26 isolates. Second, chromosomal-encoded efflux pumps (*emrR, oqxA*, and *oqxB*) genes were constitutive in all isolates. Third, gene mutations were detected in gyrase A (*gyrA*)—Ser83Phe, Asp87Aspn, and Ser83Ile—and Deoxyribonucleic acid topoisomerase IV subunit A (*parC*)—Ser80Ile (Appendix A), which are involved in DNA synthesis. A total of 29 isolates (32%, 29/89) had the tetracycline-resistance genes, *tetA* or *tetD*, while 15 isolates (17%, 15/89) carried the chloramphenicol resistance genes *cat1*, *catII*, and *catB3* (encoding chloramphenicol acetyltransferases) as well as *floR* and *cmlA* (encoding chloramphenicol efflux pumps). Resistance to sulphonamide and trimethoprim, administered as co-trimoxazole, was mediated by the *dfrA* trimethoprim resistance gene and *sul* sulphonamide resistant gene, found in 54 and 55 isolates, respectively. Resistance to other drug classes was conferred by: fosfomycins—*fosA*; macrolides—*mphA, mphE, msrE*, and *ereA2*; rifamycins—*arr2* and *arr3*; and even antiseptics—*qacEΔ1* and *qacL*.

### 3.5. Virulence Factors Associated with the K. pneumoniae Isolates

Known virulence factors involved in adherence, biofilm formation, capsule synthesis regulation, mucoid phenotype regulation, immune evasion, secretion system, serum resistance, siderophores expression (enterobactin, yersiniabactin, aerobactin, and salmochelin), efflux pump expression, allantoin utilization, and enterotoxin generation were detected among the isolates (Figure 5). The most ubiquitous were the chromosomal genes *fim, mrk*, and *ecp* for adherence and biofilm formation, which were present in all isolates except kkp043 which lacked the *fim* genes. In addition, some isolates carried multiple copies of the *mrk* genes in plasmids (Table 2). Other genes identified in all the isolates were those for serum resistance factors that determine the ‘O-antigen’ lipopolysaccharide serotype, the immune evasion factors which determine the polysaccharide capsule (K antigen) type, capsule synthesis regulation (*rcs*), efflux pump expression (*acrAB*), and enterobactin (*ent*, *fep*) (Appendix A). In addition, all isolates possessed type VI secretion system loci genes except kkp034.

This study identified no evident clustering of capsule and lipopolysaccharide types based on geographical locations or clinical presentation. There were 10 different O-loci types identified (O1, O2, O3a, O3b, O4, O5, OL101, OL103, and OL104) in the 57 assigned isolates, whereby O1 (46%, 26/57) and O2 (19%, 11/57) were the most common and clinically significant. There were 24 different K-types in the 45 assigned isolates, and the most abundant were K2 (11%, 5/45), K25 (11%, 5/45), K102 (9%, 4/45), K62 (9%, 4/45), and K24 (7%, 3/45). 

The *ybt* loci, identified in 24/89 isolates, encode for yersiniabactin siderophores found within conjugative transposons in the chromosome. Based on a new typing scheme [46] updated in the kleborate pipeline [40], this study identified four distinct *ybt* types: ybt14 found within ICEKp5, ybt15 in an ICEKp11, ybt16 in an ICEKp12, and ybt9 within an ICEKp3. In addition, the analysis revealed three isolates (Figure 5) with chromosomally encoded genes for allantoin utilization and two isolates (kkp012 and kkp045) (Figure 5) with *magA* and *K2* capsule types linked to hypervirulence [47].

Several isolates were unique in having plasmid-encoded virulence genes. For example, a hypervirulent isolate (hvKP), kkp043, was identified bearing the repB_KLEB_VIR plasmid containing *rmpA* and *rmpA2* genes, which regulate the expression of the mucoid phenotype, salmochelin (*iroBCDN*), and aerobactin (*iucABCD, iutA*). In addition, kkp083 also demonstrated an *iroBDEN* cluster and a heat-stable enterotoxin (*astA*) gene carried in an IncF(K)_1 plasmid, in contrast to the chromosomally-bound *astA* gene in kkp032. This study did not identify MDR hypervirulent isolates with AMR and hypervirulent genes [48]. Furthermore, the MDR isolates did not carry factors associated with hypervirulence, while the hypervirulent isolates were mostly antibiotic susceptible, i.e., they were non-MDR and ESBL-positive (Figure 5).

## 4. Discussion

This study characterized 89 isolates based on their clinical, geographic, genotypic, and phenotypic characteristics. It was noted that all four recognized phylogroups were represented, with *Klebsiella pneumoniae* subsp. *pneumoniae* isolates predominating, consistent with findings from other studies [49,50,51]. Some differences were observed in the characteristics of the phylogroups. For example, *Klebsiella variicola* subsp. *variicola* isolates are typically linked to bloodstream infections and UTIs [52,53]; however, those identified in this study were mainly associated with SSTIs (Figure 1) and were largely antibiotic susceptible (Figure 4). In previous studies, *Klebsiella quasipneumoniae* subsp. *similipneumoniae* isolates were linked to nosocomial infections such as UTIs [54,55]. Yet in this study, one isolate, kkp034, caused a community-acquired SSTI, potentially indicating a broader distribution of this phylogroup in Kenya than found in other countries, such as sewage in Brazil [56] and a turtle in China [57].

The lineages identified in this study were highly diverse, and most of them were local strains that have not been described in other countries. Globally disseminated high-risk lineages such as ST14, ST15, ST307, and ST607 were identified (Figure 3). These high-risk lineages are bacterial pathogens that easily acquire and disseminate antimicrobial resistance [58]. For example, ST14, ST15, and ST147 have been linked to the spread of carbapenemase resistance genes in many countries [59,60]. In this study, the two extensively drug resistant (XDR) ST147 strains carried a *bla_OXA-181_* (kkp001) and a *bla_NDM_* gene (kkp003), respectively, while the MDR ST14/15 strains carried several ESBL genes. Notably, kkp018 (ST15) harbored a mobile colistin-resistant gene, *mcr-8*, and an *AmpC* beta-lactamase gene, *bla_DHA_* (Figure 4). ST307 and ST607 are emerging strains linked to ESBL infections (Long, et al., 2017), and they were observed in six MDR strains isolated from SSTIs and UTIs (Figure 1 and 5). ST17 is a regional strain that has been implicated in outbreaks in Kilifi [6], Mwanza [61], and Kilimanjaro [62]. This study identified three MDR and ESBL positive ST17 strains, isolated from a community-acquired SSTI, a nosocomial UTI, and a hospital environmental swab (Figure 1 and Figure 5). High-risk clones have also been linked to specific serotypes, e.g., ST607-K25, responsible for a nosocomial outbreak at a neonatal intensive care unit in a hospital in France [63]. Significantly, this study identified three MDR and ESBL-positive ST607-K25 clones associated with SSTIs (Figure 5). 

The presence of these high-risk strains in Kenya indicates the significant clinical and public health threat they pose. However, we noticed that this threat was highest in Kisumu and Nairobi, the two largest cities in the country, where most of the global lineages (79%, 11/14) were identified. These cities are travel hubs with large referral hospitals serving patients from a wide geographical area locally and as well as global travelers. 

The diverse patient population could explain the concentration of local, regional and global lineages, the great strain diversity, and novel alleles in isolates from the two counties compared to Kisii, Kilifi, and Kericho Counties.

Plasmids are the main vehicle for AMR gene transmission, and in this study, we found that a majority of the AMR genes were carried in plasmids, particularly IncFIB(K) and IncFII(pKP91) (Table 2), which are common among *Enterobacterales*. The predominant plasmid replicon types belonged to the diverse Incompatibility (Inc) family [64], whose host range is mainly limited to *Enterobacterales* [65] which are known to have large multi-replicon plasmids [66]. The multi-replicon IncF plasmids contain the FII, FIA, and/or FIB replicons (Table 2 and Appendix A) which account for their high abundance (Figure 2). Unsurprisingly, antibiotic susceptible KP isolates (kkp012, kkp030, kkp102, and kkp0112) contained few or no plasmids and were ESBL negative and non-MDR. The MDR kkp070 was exceptional because although it possessed no plasmid replicons, it had several resistance genes integrated into a genetic island in the chromosome. These integration events were not uncommon, as plasmid-associated AMR genes were detected in the chromosome of several isolates: kkp005 and kkp006 carried a *bla_CTX-M-15_* gene, kkp109 had a *dfrA14* trimethoprim resistant gene, and kkp001 possessed several AMR genes (*aac(6′)-Ib9*, *bla_CTX-M-15_*, *arr-2,* and *rmtF*). The integration of *AMR* genes into the chromosome is alarming because the resistance is transferred clonally, becomes part of the core genome, and increases the spread and prevalence of non-susceptible *K. pneumoniae* lineages. 

Although the col plasmid family was the second most dominant group, the majority of col-like plasmids did not contain any resistance or virulence genes, except for the abundant colE1 type, Col (pHAD28), which typically carries *qnrS1* and other *AMR* genes [65] as observed in kkp0e21 (Table 2). In addition, col plasmids benefit *K. pneumoniae* because they produce bacteriocins lethal to rival bacteria. One of the other plasmid families identified was the pK1433 (kkp056) and is associated with the *bla_KPC2_* gene [67] that provides a mechanism for carrying and spreading KPC-type genes. The isolates with plasmids carrying a *bla_CTX-M_* gene also carried *bla_OXA_, bla_SHV_*, or *bla_TEM_* genes, implying that these resistance genes are transferred together (e.g., ESBL-positive kkp081 and kkp059 isolates possessed *bla_CTX-M-15_, bla_SHV-134_,* and *bla_TEM181_* genes). The most intriguing isolates were the ESBL-positive isolates (kkp015, kkp070, and kkp107) which only contained the *bla_SHV_* penicillinase gene. We hypothesize that the ESBL phenotypes in these isolates may have been due to alternative resistance mechanisms such as up-regulation of MDR efflux pumps. The expression of genes encoding MDR efflux pumps is noteworthy because their activity induces resistance of different antibiotic classes non-specifically and can cause phenotypic and genotypic discordance.

Examination of the virulence factors among the *K. pneumoniae* phylogroups highlighted one main difference. *K. quasipneumoniae* subsp. *similipneumoniae* did not demonstrate the Type VI protein secretion system found in the other phylogroups (Appendix A); instead, it possessed the Type II secretion as previously noted [68]. Siderophores were the most significant virulence genes detected. The ubiquitous enterobactin scavenges iron from host cells; however, their activity is neutralized by human lipocalin-2 protein [69]. In response, *K. pneumoniae* uses the more virulent yersiniabactin to bind iron and other heavy metals such as copper to avoid metal toxicity and phagocytosis using reactive oxygen species [70]. According to Holt et al. (2015), acquiring yersiniabactin is usually the first step in accumulating more potent siderophores to make them more invasive. Yersiniabactin genes were found among the *K. pneumoniae* isolates causing various clinical infections and from diverse geographical locations. Most of them were MDR and ESBL positive, making their infections persistent and antibiotic-resistant and likely contributing to their dominance over the other phylogroups.

Classical hypervirulent *K. pneumoniae* (hvKP) isolates are generally antibiotic susceptible [71], and they are characterized by having *rmpA/A2* and *magA* virulence factors as well as the *K2* capsule type [60]. This study identified one potentially hypervirulent isolate (kkp043), which carried *rmpA* and *rmpA2* genes. In addition, it also carried genes for aerobactin and salmochelin that were carried on a repB_KLEB_VIR plasmid (Table 2). A trade-off between virulence and antibiotic resistance was demonstrated in this potentially hypervirulent isolate (kkp043) as it was non-MDR and ESBL-negative and had only one other plasmid bearing the IncFIA(HI1) replicon, *dfrA5,* and *sul1* genes (Table 2). Additional examples were the two isolates (kkp012 and kkp045) with *magA* genes which were ESBL-negative and non-MDR. Furthermore, non-MDR K2 strains possessed the *ybt* operon, which the MDR K2 strains lacked (Figure 5). Despite this, there is growing concern about the rise of MDR hvKP globally [46,72,73,74,75], which cause severe infections with few treatment options [76] emphasizing the significance of monitoring virulence characteristics of *K. pneumoniae* in Kenya.

The study had several limitations. First, the two sequencing technologies utilized had different strengths and weaknesses, which could introduce bias in the genomic analysis. The draft genomes from the long reads had fewer contigs (<= 12) (Appendix A) and were more contiguous, but they had less depth/coverage, while the short reads produced less contiguous draft genomes with greater depth (>40) (Appendix A). More contiguity enabled better plasmids reconstruction, while sufficient depth enabled better assignment of multilocus sequence types. Secondly, susceptibility tests were carried out on the Vitek2^®^ platform using a limited number of antibiotics and were not verified using another phenotypic method. Nevertheless, there was good concordance between the phenotypic and genotypic results. Finally, in vivo tests were not conducted to confirm the virulence gene activity, so the virulence results are only predicted.

## 5. Conclusions

The findings of this study contribute in several ways to our understanding of the genotypic and phenotypic characteristics of Kenyan *K. pneumoniae* isolates. The multi-center approach provided more nationally relevant data, unlike prior studies with fewer isolates from single sites. These findings describe a *K. pneumoniae* population with diverse sequence types, highly abundant and diverse resistance, and virulence profiles. In addition, the presence of high-risk clones in the major cities, Nairobi and Kisumu, enhances their transmissibility within and outside the country. Further research should be conducted to correlate the genotypic findings of virulence with phenotypic data, and additional analysis should investigate the genotypic environment of the acquired antimicrobial resistance genes to determine their risk of spread.

## Figures and Tables

**Figure 1 pathogens-11-00545-f001:**
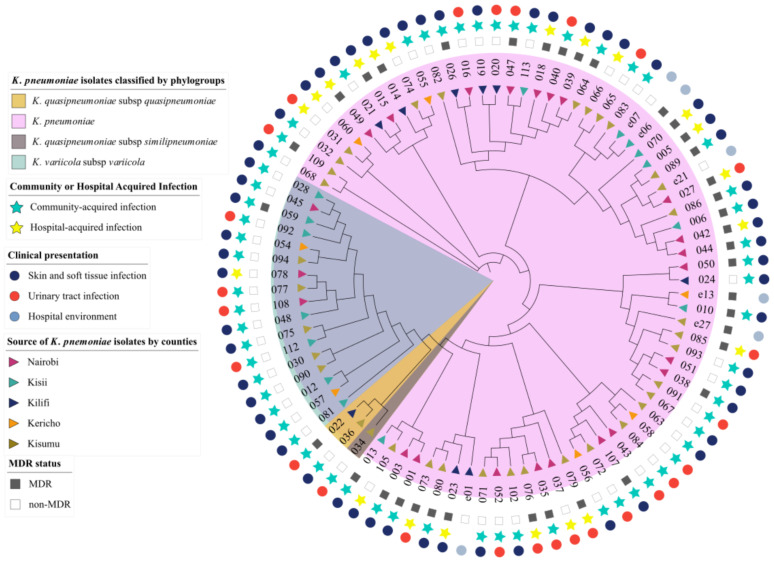
Circular cladogram showing epidemiological and clinical characteristics of *K. pneumoniae* isolates (*n* = 89). The color shading indicates the clustering of the isolates by phylogroups: *Klebsiella pneumoniae* (pink), *Klebsiella quasipneumoniae* subsp. *similipneumoniae* (brown), *Klebsiella quasipneumoniae* subsp. *quasipneumoniae* (tan), and *Klebsiella variicola* subsp. *variicola* (blue). The triangle symbols in the innermost ring represent the geographic source of the isolates: Nairobi (purple), Kisii (teal), Kilifi (blue), Kericho (orange), and Kisumu (brown). The star symbols represent the isolates recovered from community-acquired (blue) or healthcare-associated infections (yellow). The square symbols represent the multidrug resistance status of the isolates as either multidrug-resistant (black) or non-multidrug-resistant (white). The circle symbols in the outermost ring represent the clinical presentation of the isolates, i.e., skin and soft tissue infections (dark-blue), urinary tract infection (red), and the hospital environment (light-blue).

**Figure 2 pathogens-11-00545-f002:**
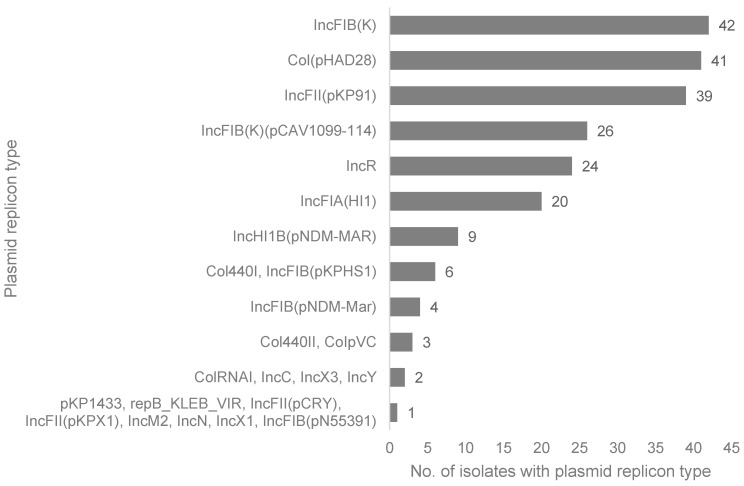
The types and number of plasmid replicons identified in the Kenyan *K. pneumoniae* isolates (*n* = 89).

**Figure 3 pathogens-11-00545-f003:**
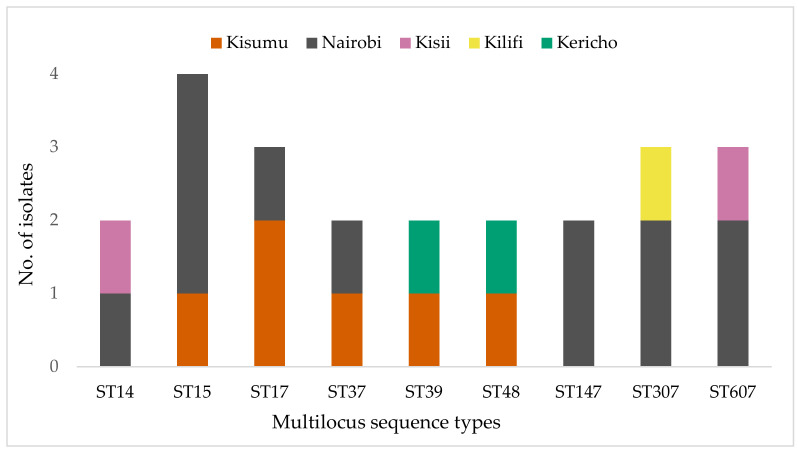
Distribution of the most abundant sequence types of the *K. pneumoniae* isolates by county.

**Figure 4 pathogens-11-00545-f004:**
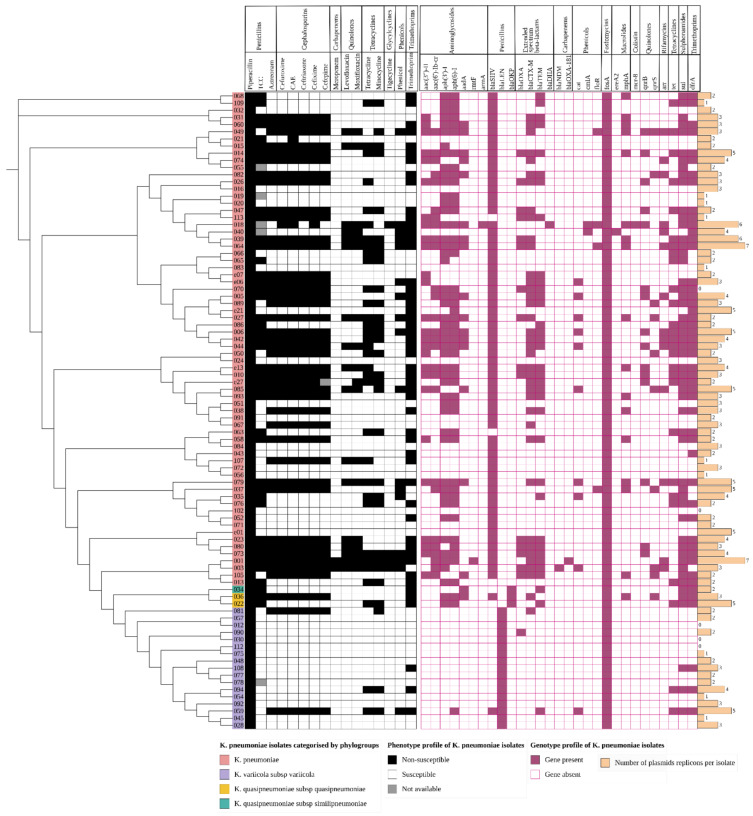
A heatmap of the phenotypic and genotypic profiles and plasmid replicon abundance of *K. pneumoniae* isolates (*n* = 89). The cladogram on the left with colored labels on the edge indicates the clustering of the isolates by phylogroups: *K pneumoniae* (pink), *K. quasipneumoniae* subsp. *similipneumoniae* (teal), *K. quasipneumoniae* subsp. *quasipneumoniae* (gold), and *K. variicola* subsp. *variicola* (grey). The phenotypic profile is represented as an isolate being non-susceptible (black) or susceptible (white) to the antibiotic indicated on the bottom column header (and the drug class it belongs to on the top column header). The grey square indicates an isolate whose phenotypic result was not available. The genotypic profile is represented as a gene present (purple) or absent (white with purple outline). The genes are indicated on the column header and the drug class on the top column header. The beige bar plot represents the number of plasmid replicons found in the isolates; CAE—cefuroxime axetil; TCC—ticarcillin/clavulanate.

**Figure 5 pathogens-11-00545-f005:**
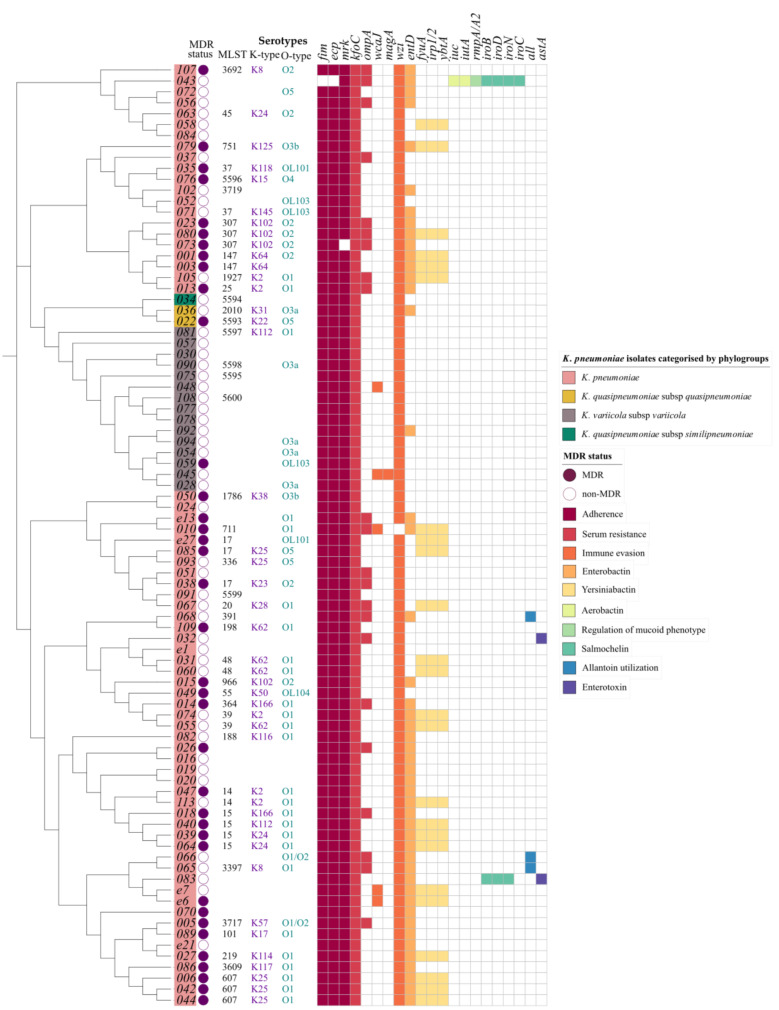
A heatmap of the multidrug resistance status, ST, serotype, and virulence gene content of *K. pneumoniae* isolates (*n* = 89). The cladogram on the left with colored labels on the edge indicates the clustering of the isolates by phylogroups: *K. pneumoniae* (pink), *K. quasipneumoniae* subsp. *similipneumoniae* (blue), *K. quasipneumoniae* subsp. *quasipneumoniae* (gold), and *K. variicola* subsp. *variicola* (green). The circular symbols represent the multidrug resistance status of the isolates as either multidrug-resistant (purple) or non-multidrug-resistant (white). The serotypes assigned to the isolates are indicated as follows: multilocus sequence type (MLST) (black), capsule type (purple), and O type (teal). The virulence profile is represented as a gene present (color) or absent (white) for factors: adherence (purple), serum resistance (blue), immune evasion (pink), enterobactin (green), yersiniabactin, aerobactin, salmochelin, regulation of mucoid phenotype, enterotoxin and allantoin utilization; blank spaces indicates unassigned serotypes.

**Table 1 pathogens-11-00545-t001:** Allelic profiles of *K. pneumoniae* isolates with novel multilocus sequence types.

Isolate ID	BIGSdb ID	Species	MLST	*gapA*	*infB*	*mdh*	*pgi*	*phoE*	*rpoB*	*tonB*
kkp022	16011	*K. quasipneumoniae*	5593	17	19	39	39	552	21	262
kkp034	16017	*K. quasipneumoniae*	5594	18	22	56	162	556	13	51
kkp075	16032	*K. variicola*	5595	45	18	21	105	455	22	786
kkp076	16033	*K. pneumoniae*	5596	2	9	2	1	13	1	787
kkp081	16037	*K. variicola*	5597	16	24	21	40	106	17	67
kkp090	16041	*K. variicola*	5598	16	18	21	33	55	17	341
kkp091	16042	*K. pneumoniae*	5599	294	3	1	1	4	331	4
kkp108	16051	*K. variicola*	5600	306	24	21	27	47	22	188

ID—identity; MLST—multilocus sequence type, the seven housekeeping genes for *K. pneumoniae* strain typing (*gapA*—glyceraldehyde 3-phosphate dehydrogenase, *infB*—translation initiation factor 2, *mdh*—malate dehydrogenase, *pgi*—phosphoglucose isomerase, *phoE*—phosphoporine E, *rpoB*—beta-subunit of Ribonucleic acid polymerase B, *tonB*—periplasmic energy transducer (Diancourt, 2005); BIGSdb—*K. pneumoniae* MLST database.

**Table 2 pathogens-11-00545-t002:** Antimicrobial resistance and virulence genes identified in the plasmids of the *K. pneumoniae* isolates; ID—identity.

Isolate ID	Plasmid ID	Plasmid Replicon	AMR Genes	Virulence Genes
kkp001	kkp001_p002	IncFII(pKPX1), IncR	*aph(3″)-ib, aph(6)-id, CTX-M-15, OXA-181, TEM-181, dfrA, sul2*	
kkp005	kkp005_p002	IncFIA(HI1),IncR	*aac(6′)-ib-cr, aph(3″)-ib, aph(6)-id, CTX-M-15, qnrB, TEM-181, aadA, arr3, CatII, dfrA, qacEdelta1, sul1, sul2*	
kkp006	kkp006_p003	IncFIA(HI1), IncR	*aac(6′)-ib-cr, aph(3″)-ib, aph(6)-id, qnrB, aadA, arr3, CatII, dfrA, qacEdelta1, sul1, sul2, TetD*	
kkp019	kkp019_p001	IncFIA(HI1)	*aph(3″)-ib, aph(6)-id*	*mrkABCDFJ*
kkp020	kkp020_p001	IncFIA(HI1)	*aph(3″)-ib, aph(6)-id*	*mrkABCDFJ*
kkp024	kkp024_p002	IncFIA(HI1)		*mrkABCDFJ*
kkp026	kkp026_p001	IncFIB(K), IncFII(pKP91)	*aac(3)-iie, aac(6′)-ib-cr, aph(3″)-ib, aph(6)-id, CTX-M-15, OXA-1, qnrB, TEM-181, dfrA, sul2, TetA*	
kkp032	kkp032_p001	IncFIB(K)	*aph(3″)-ib, aph(6)-id*	*mrkABCDFJ*
kkp034	kkp034_p002	IncY	*TEM-181, aadA, dfrA, qacEdelta1, sul1, sul2*	
kkp036	kkp036_p002	IncN	*SHV-134*	
kkp037	kkp037_p001	IncC, IncFIB(K), IncFII(pKP91)	*aac(6′)-ib, aph(3″)-ib, aph(3′)-ia, aph(6)-id, floR, mphA, qacEdelta1, sul1, sul2*	
kkp037	kkp037_p002	IncX3	*qnrS, SHV-134*	
kkp039	kkp039_p001	IncFIB(K), IncFII(pKP91)	*aadA, dfrA, mphA, qacEdelta1, sul1*	
kkp043	kkp043_p001	repB_KLEB_VIR		*iroBCDN, iucABCD, iutA, rmpA, rmpA2*
kkp043	kkp043_p002	IncFIA(HI1)	*dfrA, qacEdelta1, sul1*	
kkp051	kkp051_p001	IncFIB(K)(pCAV1099-114), Col(pHAD28), IncR	*aph(3″)-ib, aph(6)-id*	
kkp052	kkp052_p003	IncR	*aph(3″)-ib, aph(6)-id, dfrA, qacEdelta1, sul1, sul2*	*mrkABCDF*
kkp066	kkp066_p001	IncFIB(K), IncFII(pKP91)	*aph(3″)-ib, aph(6)-id, SHV-120, sul2, TetD*	
kkp068	kkp068_p001	IncFIB(K), IncFII(pKP91)	*aph(3″)-ib, aph(6)-id, TEM-181, dfrA, mphA, sul2*	
kkp083	kkp083_p001	IncFIB(K)		*astA, iroBDEN*
kkp090	kkp090_p001	IncFIB(K), IncFII(pKP91)	*OXA-926*	
kkp094	kkp094_p002	IncFIA(HI1)	*ant(3″)-iia, dfrA, qacEdelta1, sul1, TetA*	
kkp108	kkp108_p002	IncFII(pKP91)	*dfrA, qacEdelta1, sul1*	
kkp0e13	kkp0e13_p001	IncFII(pKP91)	*aac(3)-iie, aac(6′)-ib-cr, aph(3″)-ib, aph(6)-id, CTX-M-15, OXA-1, qnrB, TEM-181, dfrA, sul2, TetA*	
kkp0e13	kkp0e13_p002	IncM2	*aac(3)-IId, CTX-M-15, TEM-181, mphA*	
kkp0e21	kkp0e21_p004	Col(pHAD28)	*aph(6)-id, dfrA*	
kkp0e27	kkp0e27_p001	IncFIB(K), IncFII(pKP91)	*aph(3″)-ib, aph(6)-id, CTX-M-15, qnrB, TEM-181, dfrA, sul2, TetA*	
kkp0e7	kkp0e7_p001	IncFIB(K), IncFII(pKP91)	*aac(3)-iie, CTX-M-15, TEM-181*	

## Data Availability

All data generated or analyzed during this study are included in this published article. All genome assemblies are deposited in the NCBI GenBank database under BioProject PRJNA777842. Eight novel allelic profiles were assigned and deposited in the *K. pneumoniae* MLST database https://bigsdb.web.pasteur.fr/ (accessed on 25 May 2021).

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
