# Peer review of "Antimicrobial Resistance and Virulence Characteristics of Klebsiella pneumoniae Isolates in Kenya by Whole-Genome Sequencing"

_pathogens, 2022, doi:10.3390/pathogens11050545_

Round 1
Reviewer 1 Report
Comments: Pathogens-1676220
Antimicrobial resistance and virulence characteristics of Klebsiella pneumoniae isolates in Kenya
The authors of this study investigated the epidemiology and distribution of resistance and virulence genes in clinical isolates of K. pneumoniae from five counties in Kenya using whole-genome sequencing and bioinformatics. As antibiotic resistance is increasingly becoming one of the top 10 global health threats to humanity, this study may have public health significance. The paper is generally well written, but I have listed some general and some specific issues. The points probably all involve only edits to the text; however, although they do relate to some important aspects that, once addressed, could significantly improve the paper's utility.
Major issues/comments
The authors of the study used two different sequencing platforms to perform WGS of the bacterial isolates, and this was acknowledged by the authors as one of the limitations of the study. Since two different sequencing platforms were used, this may affect the data output and subsequently the results and interpretation. How did the authors ensure that the sequencing platforms did not introduce bias into the results? To ensure that there was no bias in the results, did the authors attempt to analyze the data obtained from the two sequencing platforms independently? Regarding the selection of samples used for the two platforms: Are they randomly selected or is there some kind of logic is used to consider the bacterial isolates for the sequencing platforms?
It is also good to validate the data generated by both platforms independently and together with additional bioinformatics tools such as Core Genome cgMLST and Whole genome-SNP variant calling for epidemiological investigation, as described in the literature below. These high-resolution cgMLST/SNP analyses can provide interpretation that provides evidence of nosocomial transmission and compare these results to isolates we have locally with national findings, thereby clarifying transmission pathways.
“One means of exploiting WGS data is the identification of single nucleotide polymorphisms (SNPs) that vary among isolates. SNPs can be highly informative markers, which are capable of revealing evolutionary histories of homogenous groups (Octavia and Lan, 2010) and detecting and tracing outbreaks” (https://doi.org/10.1016/j.ijfoodmicro.2018.02.02) “cgMLST is a viable alternative high-resolution analysis approach, which is highly reproducible and scalable. Furthermore, cgMLST can be readily implemented in laboratories that only have access to web-based bioinformatics analysis tools, which makes it of particular utility in the resolution of multi-country disease outbreaks.” (https://doi.org/10.1016/j.ijfoodmicro.2018.02.02).
Minor comments/suggestions
It may be useful to have a bar graph showing the timelines of bacterial isolates collected over the period of 5 years.
Abstract:
Line #20: “whole genome sequencing” needs to be written as whole-genome sequencing (WGS) and needs to be consistent thought the manuscript.
Introduction:
Line #46 and 47: “including Klebsiella pneumoniae subsp. pneumoniae, Klebsiella quasipneumoniae subsp. quasip-neumoniae, Klebsiella quasipneumoniae subsp. similipneumoniae and Klebsiella variicola subsp. variicola [2]”. In the article (Wyres et al., 2020) there is no “Klebsiella pneumoniae subsp. pneumoniae” its only Klebsiella pneumoniae.
Additionally, there are seven K. subspecies: Klebsiella pneumoniae (Kp1) • Klebsiella quasipneumoniae subsp. quasipneumoniae (Kp2), Klebsiella variicola subsp. variicola (Kp3), Klebsiella quasipneumoniae subsp. similipneumoniae (Kp4), Klebsiella variicola subsp. tropica (Kp5), Klebsiella quasivariicola (Kp6), and Klebsiella africana (Kp7) subspecies instead of only 4 have been mentioned in the manuscript.
Figures:
In figure legends and titles, the bacteria names are not italicized and needs to be corrected.
Author Response
How did the authors ensure that the sequencing platforms did not introduce bias into the results?
The two sequencing platforms have different advantages and disadvantages. For example, the MLSTs were better assigned by the Illumina reads due to the higher sequence read accuracy whereas the plasmid profiles were more complete using the Nanopore technology due to the longer read lengths. However the core results of the manuscript i.e. the AMR and virulence genes were not significantly biased by the different technologies.
To ensure that there was no bias in the results, did the authors attempt to analyze the data obtained from the two sequencing platforms independently?
Each isolate was sequenced on only a single platform. The raw reads for each platform were assembled using different tools, but to eliminate bias, all the assemblies were processed using similar tools for the reported outcomes.
Regarding the selection of samples used for the two platforms: Are they randomly selected or is there some kind of logic is used to consider the bacterial isolates for the sequencing platforms?
The isolates used in this study were obtained from a surveillance study which routinely sequences multidrug resistant isolates on an Illumina Miseq platform. The Nanopore long-read sequencing method was used to sequence additional MDR and non-MDR isolates to better resolve full length plasmid sequences (Table 2), which is not possible with the short reads, as mentioned in line 293 - 294.
It is also good to validate the data generated by both platforms independently and together with additional bioinformatics tools such as Core Genome cgMLST and Whole genome-SNP variant calling for epidemiological investigation, as described in the literature below. These high-resolution cgMLST/SNP analyses can provide interpretation that provides evidence of nosocomial transmission and compare these results to isolates we have locally with national findings, thereby clarifying transmission pathways...
Thank you for this suggestion. Analyzing the transmission and evolution of strains will definitely provide more information on how to control and monitor antimicrobial resistance. This approach is currently being taken and will be addressed in a subsequent paper focused on epidemiology and transmission. It falls outside of the scope of this paper which was limited to reporting on the AMR and virulence genes, MLST and plasmid profiles from the study isolates. One limitation to performing cgMLST analysis is that the only open source tool CLC Genomics does not include K. pneumoniae.
It may be useful to have a bar graph showing the timelines of bacterial isolates collected over the period of 5 years.
We appreciate this suggestion. The timelines of bacterial isolates will be relevant in a subsequent paper focused on epidemiology and transmission and is outside the scope of this manuscript.
Line #20: “whole genome sequencing” needs to be written as whole-genome sequencing (WGS) and needs to be consistent thought-out the manuscript
All occurrences of “whole genome sequencing” have been hyphenated.
Line #46 and 47: “including Klebsiella pneumoniae subsp. pneumoniae, Klebsiella quasipneumoniae subsp. quasip-neumoniae, Klebsiella quasipneumoniae subsp. similipneumoniae and Klebsiella variicola subsp. variicola [2]”. In the article (Wyres et al., 2020) there is no “Klebsiella pneumoniae subsp. pneumoniae” its only Klebsiella pneumoniae.
All the occurrences of “Klebsiella pneumoniae subsp. pneumoniae” have been changed to “Klebsiella pneumoniae”
Additionally, there are seven Klebsiella subspecies: Klebsiella pneumoniae (Kp1) • Klebsiella quasipneumoniae subsp. quasipneumoniae (Kp2), Klebsiella variicola subsp. variicola (Kp3), Klebsiella quasipneumoniae subsp. similipneumoniae (Kp4), Klebsiella variicola subsp. tropica (Kp5), Klebsiella quasivariicola (Kp6), and Klebsiella africana (Kp7) subspecies instead of only 4 have been mentioned in the manuscript.
The statement in lines 46 – 49 has been changed to include the three species that were omitted.
Figures: In figure legends and titles, the bacteria names are not italicized and needs to be corrected.
The bacteria names have been italicized.
Reviewer 2 Report
Dear Authors,
The manuscript “Antimicrobial resistance and virulence characteristics of Klebsiella pneumoniae isolates in Kenya” written by Muraya et al. is well written and the data are presented in an easy to understand and interpret manner.
The title, research goals, result, and final findings correspond well.
Below I present my minor, mainly editorial concerns with suggestions for improvement.
Title: I suggest to add […] in Kenya identified by Whole-Genome Sequencing
Line 7: add coma before Nairobi.
Line 41, and throughout the manuscript: do not use KP for Klebsiella pneumoniae, instead of KP write K. pneumoniae.
Line 54: change [5]-[7] to [5-7].
Line 57, and throughout the manuscript: Enterobacteriales and other Latin names - species,
genera, etc., should be written in italics.
Line 54: change [6],[10] to [6, 10].
Line 81: add the reference number Musila et al. (2021)
Line 97: change [6],[13] to [6, 13].
Introduction section:
I recommend adding a few sentences about the epidemiological/microbiological
screening for MDROs among patients in Kenya.
Line 115: change K. pneumoniae Species Complex to K. pneumoniae complex
Section 2.3
Line 156-157: please, provide information about the assembling errors and the percentage of the genome sequencing coverage.
What method has been used for these strains? Illumina or MinION?
Your research would be more valuable if you performed an ST typing for these 38 strains by another method.
Line 158: […](STs) of the KP, change to […](STs) of the 51 K. pneumoniae […]
Line 159: You cannot state that ST15 was 4/89, and […] since 38 isolates were not assigned to any ST type!!!
Line 197 and throughout the manuscript: gene symbols should be italicized
Line 205 and throughout the manuscript: eg. blaOKP, should be written as “bla” and OKP in a
subscript.
Line 440: CLSI guidelines – add the reference.
Line 446: what was the growth medium for the remaining strains?
References:
The citation style must be standardized according to the publisher's requirements,
e.g. [5] and [6], they are given in 2 different styles
The materials and methods section should be placed after the Introduction section.
Overall, I recommend accepting the manuscript by Muraya et al. for publication in the
Pathogens (doi.org/10.3390/….) after the application of all the necessary corrections.
Yours sincerely,
Reviewer
Author Response
Title: I suggest to add […] in Kenya identified by whole-genome sequencing
The title has been modified to include the method of analysis as suggested.
Line 7: add coma before Nairobi.
The comma has been added before Nairobi
Line 41, and throughout the manuscript: do not use KP for Klebsiella pneumoniae, instead of KP write K. pneumoniae. Line 57, and throughout the manuscript: Enterobacterales and other Latin names - species, genera, etc., should be written in italics.
KP has been changed to K. pneumoniae. The bacteria species and genera names have been italicized.
Line 54: change [5]-[7] to [5-7]. Line 54: change [6], [10] to [6, 10]. Line 97: change [6], [13] to [6, 13].
The reference citation style has been changed to Vancouver and this has resolved the reference numbering differences.
Line 81: add the reference number Musila et al. (2021)
The reference citation has been added (line 82)
Introduction section: I recommend adding a few sentences about the epidemiological/microbiological screening for MDROs among patients in Kenya.
Additional details on the epidemiological and microbiological screening for MDROs among patients in Kenya have been added in lines 69 – 75 and 105 – 113.
Line 115: change K. pneumoniae Species Complex to K. pneumoniae complex
K. pneumoniae Species Complex has been changed to K. pneumoniae Complex in Line 200 as suggested.
Section 2.3 Line 156-157: please, provide information about the assembling errors and the percentage of the genome sequencing coverage.
The genome coverage data has been provided in the Supplementary Table S1 which has been cited in line 243. The statement on assembly errors has been removed because these errors are speculative and not quantifiable. The statement in line 243 now reads “due to low coverage (Supplementary table S1)”.
What method has been used for these strains? Illumina or MinION?
Both Illumina and Minion technologies were used for sequencing as indicated in Section 2.3.
Your research would be more valuable if you performed an ST typing for these 38 strains by another method.
We concur with this recommendation however this would require re-analysis of these strains using a higher fidelity sequencing method or higher coverage. Nanopore sequencing which cannot be conducted now due to financial constraints. This work is planned and the sequences and STs updated in the genomic databases.
Line 158: […] (STs) of the KP, change to […] (STs) of the 51 K. pneumoniae […]
All the occurrences of KP have been changed to K. pneumoniae.
Line 159: You cannot state that ST15 was 4/89, and […] since 38 isolates were not assigned to any ST type!!!
Thank you for pointing out this error. There were thirty seven unassigned isolates so the denominator has been changed from 89 to the 52 which were assigned STs.
Line 197 and throughout the manuscript: gene symbols should be italicized Line 205 and throughout the manuscript: eg. blaOKP, should be written as “bla” and OKP in a subscript.
The gene names have been italicized and subscripted, where applicable.
Line 440: CLSI guidelines – add the reference.
The reference (Weinstein MP., 2018) has been added in line 146.
Line 446: what was the growth medium for the remaining strains?
The whole genome sequence data of the 40 isolates were already available and were not cultured again for this study from a parent study which has been referenced in this paper. The paragraph has been edited to clarify this point.
The citation style must be standardized according to the publisher's requirements, e.g. [5] and [6], they are given in 2 different styles
The reference citation style has been changed to Vancouver which has resolved the issue.
The materials and methods section should be placed after the Introduction section.
The materials and methods section has been moved to after the Introduction section as suggested.